# Content Validity of a New Soccer (Football) Return-to-Play Test: The RONDO-TEST

**DOI:** 10.3390/jfmk10010003

**Published:** 2024-12-25

**Authors:** Sergi Matas, Carlos Lalín, Francisco Corbi, Antoni Planas-Anzano, José M. Moya, Sebastià Mas-Alòs, Xavier Peirau-Terés

**Affiliations:** 1Institut Nacional d’Educació Física de Catalunya (INEFC), Partida la Caparrella, 97, E-25192 Lleida, Spain; fcorbi@gencat.cat (F.C.); aplanas@gencat.cat (A.P.-A.); xpeirau@gencat.cat (X.P.-T.); 2Human Movement Research Group (GRMH), University of Lleida (UdL), [2021 SGR 01619], E-25002 Lleida, Spain; 3Tottenham Hotspur F. C., White Hart Lane, Bill Nicholson Way, 748 High Road, Tottenham N17 0AP, UK; carloslalinovoa@yahoo.es; 4Facultad de Formación del Profesorado y la Educación, Departamento Educación Física, Deporte y Motricidad Humana, Autonomous University of Madrid, Ciudad Universitaria de Cantoblanco CA, E-28049 Madrid, Spain; josemaria.moya@uam.es

**Keywords:** clinical decision-making, Delphi technique, functional test, leg injuries, sport, field test, athletic performance, agility, health, team sports

## Abstract

**Featured Application:**

**The RONDO-TEST is a low-cost and soccer (football)-specific tool used to evaluate players’ skills in the field in relation to potential injury risk factors.**

**Abstract:**

Objectives: The aim was to assess the content validity of a new field test on general and soccer-specific motor skills before return to play. Methods: The RONDO-TEST was assessed by a Delphi panel for its content validity. It included a survey to evaluate 16 items related to the test consisting of four 10 m lines which cross over at their mid-point, resulting in eight 5 m sectors that include locomotor skills (speeding, moving sideways, side cutting, and jumping) and soccer-specific technical skills (dribbling, slalom course, and kicking/passing). The content validity was calculated with the Aiken’s V coefficient of acceptance at 0.69 and 95% of confidence interval. Results: Eight experts participated in the Delphi and agreed on the administration procedures after three rounds of suggestions. Major changes included the order of execution and the descriptions of the sectors. The results showed consensus (V = 1, maximum acceptance) for the clarity of instructions, the relevance of the skills to be evaluated, the order of execution, the materials and the relevance of measuring total and partial sector times. Conclusions: The RONDO-TEST may be feasible and simple to administer and evaluate technical functional skills (actions) and condition-related abilities (e.g., the ability to repeat the test, fatigue curve, etc.), which are relevant aspects for return to play under optimum conditions.

## 1. Introduction

Football or soccer is a team sport characterized by continuous and intermittent high-intensity actions [1], combined with sport-specific motor and technical skills, such as passing, changing direction, kicking, jumping, moving, and dribbling with an environment of uncertainty [2,3]. These features, coupled with other facilitating factors (e.g., individual fitness and skills), predispose players to injuries, and consequently, soccer practice is considered a high-risk activity [4,5]. Several studies established an injury incidence in adult players to be between 10 and 35 injuries per 1000 h of play [5,6]. Injuries in soccer matches are higher than in other team sports; there are 2.4 injuries per match (95% CI, 2.0–2.8) or 108 for every 1000 player matches (95% CI, 89–127), compared to 1.6 and 0.65 injuries per match in handball and basketball, respectively [7]. Given the high social, sporting, and economic repercussions of injury, especially at highly competitive levels, it is considered essential to anticipate injury and return to practice as quickly as possible after injury recovery in an optimal physical and technical condition [8,9].

The return-to-play (RTP) process and the final decision are complex due to their multifactorial nature [10]. It should be highlighted that RTP as a process in clinical practice may refer from full return without restrictions to allow exercising, but the final decision after the process includes the medical clearance of the athlete for full participation without restriction [10]. Medical and injury risk factors to consider before an RTP decision include the complete suppression of pain, repair of injured tissues, restoration of metabolic pathways that guarantee energy supply, elimination of muscle and joint disbalance due to inactivity, and re-education of general and soccer-specific movement patterns [11]. These risk factors are essential to consider because the risk of re-injury increases when rehabilitation is inadequate or the RTP is premature [12,13].

Fitness tests were used to facilitate decision-making (e.g., Hop Test, Triple-Hop Test, T-Test, Illinois Test, Multi-Stage Fitness Test, Cooper Test, Shuttle Test, and Balance Test) regarding an RTP. Validation studies exist for the beforementioned tests—Hop Tests [14], Illinois Tests [15], Shuttle Tests [16], and Balance Tests [17]—but the prognostic capacity following serious injuries is yet unclear [14]. On-field RTP programs were validated for specific muscle injuries in soccer, but they lack validated objective measures for motor skills to assist decision-making [18,19]. Various studies [20,21] identified other means of evaluating physical fitness or movement capacities for healthy and injured soccer players, including an ecological selection of tests (clinical and in-field, physical and psychological) after anterior cruciate ligaments [22]. However, the previous tests do not comprehensively evaluate basic motor skills, soccer-specific technical skills, and injury risk patterns. Moreover, the application of a multitude of tests is unfeasible in real practice due to limited time and increased fatigue and risk of injury. For example, validity studies of laboratory testing [23] may not be feasible broadly (e.g., grassroot or regional sport clubs). Thus, objective in-field tests are needed to integrate conditional skills and patterns, based on the functional requirements of each soccer-specific action, which are as close as possible to competitive performance, to ensure the safety of athletes during implementation.

As suggested by Herring et al. [11], there is a need to control the evolution of players during the re-adaptation and re-training process and to guide the coaching staff during the re-integration process. Additionally, Lalín and Peirau [8] suggested that functional tests assessing motor skills and specific soccer patterns during the re-adaptation and re-training process permit the assessment and identification of injury risk factors, aiding the final decision of RTP. Therefore, the main aim of the present study was to design and assess the content validity of a new field test to assess general and soccer-specific motor skills aimed at identifying relevant assessment before RTP, namely the RONDO-TEST.

## 2. Materials and Methods

### 2.1. Test Design

The RONDO-TEST was inspired by the ‘rondo’, a common specific drill in soccer training, in which fundamental movement skills (e.g., side cutting, accelerating, stopping, and jumping) [24,25,26] are combined with soccer-specific technical skills (e.g., kicking and dribbling) [3,25,27] that coincide with the most common non-contact injury mechanisms in soccer (see Table 1). The RONDO-TEST includes four 10 m lines which cross over at their mid-point, resulting in eight 5 m sectors that have to be executed as fast as possible (see Figure 1). Therefore, the RONDO-TEST provides a more comprehensive evaluation of performance in the context of soccer unlike tests that assess a single sports action in an isolated and decontextualized environment.

### 2.2. Delphi Procedures

The Delphi technique is a mixed-methods research approach, designed for the collection and aggregation of informed judgements, relative to a key issue or concept. Through a series of rounds, the Delphi provides a consensus framework from a recognized panel of experts [28]. The first RONDO-TEST was drafted and submitted to a panel of eight experts with broad experience in sports performance: two professional soccer team physicians, two UEFA pro-level soccer coaches, two professional soccer team sport physiotherapists, one professor of biomechanics, and one athletics coach who was also an engineer. The Delphi technique was used to overcome the biases and limitations of the test designers and provide different perspectives to achieve a consensus, based on intersubjective judgement [29]. Panel experts were informed of the study aims and methodology. Experts received by email the RONDO-TEST description and were asked to answer a survey with 16 items that included a 5-point Likert scale (5—strongly agree, 4—agree, 3—no opinion, 2—disagree, and 1—strongly disagree) and open questions to clarify the answers. Items were the following: (A1) clarity of instructions to perform the test; (B2) order of execution and global development; (C3) material organization by regards of type and distance, (D4-D11) sector description and appropriateness of the skill (Segments 1 to 8); (E12-E15) relevance to measure total time, partial time per each sector, partial time per groups of sectors (i.e., skills with the ball and skills without the ball), (F16) appropriateness and feasibility of using photoelectric beam and acoustic signal to record time. One of the authors (SM) acted as a contact person for coordinating the Delphi panel actions (i.e., sending and receiving communications by email, organizing, and summarizing all comments provided by the experts, and clearing doubts), and processing the data following average scores of the survey: 1–2.9, major changes; 3–3.9, minor changes; 4–5, acceptance. A subsequent round of survey distribution with a description of changes was administrated until a consensus was achieved on the final scores. The Aiken’s V coefficient value was calculated for the RONDO-TEST items content relevance. The coefficient was obtained each round with a critical reference value of the coefficient of acceptance at 0.69 and a 95% score confidence interval [30,31,32]. Data were processed using MS Excel© 2016.

This study was conducted according to the guidelines of the Declaration of Helsinki [33] and approved by the Clinical Research Ethics Committee of the Sports Administration of Catalonia, Catalan Sports Council, Government of Catalonia (Approval No 05_2018_CEICGC). Consent to use the images of the soccer player in Appendix A was specifically obtained.

## 3. Results

The Delphi panel of experts provided three rounds of suggestions. (See Table 2). The results from the first round showed disagreement (V < 0.69) for the description and instructions of the overall RONDO-TEST (A1) and instructions for each segment (D4-D11), the order of execution of the skills (B2), and the organization of the material (C3), whereas the relevance to measure total and partial times (E12–E15) and the use of the material (E16) to record time obtained consensus (V = 1). Consequently, RONDO-TEST was adapted according to the suggestions, that is, the order of execution from ‘L’ turns (see Figure 1 and Appendix A changed to full lines covering two segments crossing the midpoint, the description of how to execute the sectors was reworded, and a second round of the survey was distributed to the Delphi panel with a description and justification of the changes according to the experts’ suggestions, maintaining anonymity to other experts.

The results from the second round showed disagreement (V < 0.69) for the clarity of the instructions to perform the test (A1) and descriptions for two sectors (D10, D11). Again, RONDO-TEST instructions were reworded according to the suggestions, and the survey was sent for the third round, which resulted in a broad consensus with a V = 1 in all items.

Relevant contributions were made with respect to the sequence order of sectors (related to basic or soccer-specific skills) and the importance of collecting partial sectoral time measurements, arguing that sectoral time evaluation may provide information on selected specific skills that are useful for biomechanical, physiological, or psychological demands of a certain injured profile.

It was agreed that the first four sectors assess fundamental locomotor skills [24] and the other four technical skills [26]. The players had to run off four segments without the ball and four segments with the ball following the indications provided; otherwise, in case of unsuccessful attempt, the trial would be null. The test should be administrated on a natural grass or artificial turf pitch surface based on the players regular practice, and participants should perform with their regular footwear (i.e., soccer boots with aluminum or rubber cleats). The main outcomes were the total and sectional times to perform the test. It was agreed that the execution time would be measured with a sensitivity of cent seconds, as athletic runs are, and no extra time would be added due to penalties. The setting up the RONDO-TEST by a single test administrator lasted 10–15 min (see Appendix A, a video showing the test set up; see Figure 2 for a complete description of the RONDO-TEST; see Appendix A for detailed procedures for development; see Appendix A, a video which demonstrates the RONDO-TEST execution).

## 4. Discussion

The main objective of this study was to design and assess the content validity of the RONDO-TEST as a novel functional test to evaluate basic and soccer-specific skills. This study initiated a meaningful discussion among various experts in relation to basic and specific skills to be measured by a feasible test by means of equipment. The selection of tasks to evaluate skills were appropriate, and the sequence followed basic to specific skills in soccer. The minimum level of Aiken’s V coefficient can be considered acceptable from a value of 0.50 for the validation of observation instruments [34]. Other authors suggested Aiken’s V coefficient value of 0.70 as more suitable for social science studies [30,35]. The result obtained for the RONDO-TEST was the maximum (V = 1) with a confidence level of 95%, which is in line with the standard set out elsewhere [36].

The RONDO-TEST is a simple tool that enables most motor patterns performed in soccer to be assessed jointly using a single test. It is the first approach to developing a new functional test to provide information about soccer-specific functional patterns. Two considerations should be noted. First, performing this test involves the repetition of a previously defined circuit, which practically eliminates any potential influence of the uncertainty of the environment. Second, situations of cooperation or opposition increase the level of competition and, consequently, the risk of injury [37], but the RONDO-TEST does not contain any such situations. Several studies found that injury rates in soccer are higher in competition matches than in training [38], and the influence of the cooperation and opposition component might account for that fact, at least in part. Most of the injuries occurring in soccer happen in contact situations, with the presence of chaos, fatigue, and neurocognitive management. Although this could be considered a limitation, we believe that this test should be considered a preliminary step towards because the usefulness of fitness and skill tests to predict the onset of injuries and the RTP decision is controversial [39]. RONDO-TEST procedures received consensus despite not including decision-making during the execution (e.g., visual stimuli to decide the segment to perform), as it was designed originally. The reason for it is to administer the RONDO-TEST with the very same conditions, so the main outcomes (total and partial times) will not be affected by decision-making as a confounding variable. RTP is different depending on the type of injury (i.e., injury mechanism and tissue being affected), and the RONDO-TEST can lead to obtaining basal values prior the onset of injury and, hence, monitor basic and soccer-specific motor patterns alongside the RTP process thanks to the RONDO-TEST being segmented. Qualitative information may be observed if the execution of the RONDO-TEST is video-recorded, as for analyzing possible reasons of poor execution. Some tests used to predict the onset of injuries in soccer include the Landing Error Scoring System (LESS), which identifies ACL injury risks during landing in soccer [40]; the Star Excursion Balance Test (SEBT), which assesses lower limb neuromuscular imbalances [41]; and the Isokinetic Strength Ratio, which assesses agonist and antagonist muscle imbalances between hamstrings and quadriceps (H: Q ratio %) [42]. The aforementioned tests analyze a single movement pattern alone, whereas injury mechanisms likely include several movement patterns found in soccer. Moreover, these tests are time-consuming, not specific to soccer, and are usually performed away from the soccer pitch.

## 5. Conclusions

The design of RONDO-TEST includes regular situations in soccer such as keeping the ball in small-sized areas, which is common practice in training because of its transference to situation in matches, the ‘rondo’ exercises. Physical and technical skills required to perform ‘rondo’ satisfactorily, that is, without uncertainty, which inspired us to name the test, the RONDO-TEST. This test may provide a feasible and simple way to objectively evaluate soccer motor skills and, potentially, the readiness of soccer players to participate in team trainings after an injury and, therefore, lowering the risk of re-injury because of premature return to play.

Future studies should address the concurrent or convergent validity with selected equipment, such as photoelectric beam cells and specific software, and specific protocols for test administration. Also, reference values from different populations (according to sex, age, level of performance) may provide the assessment of technical functional skills (actions) and condition-related abilities (the ability to repeat the test, fatigue curve, etc.), which are relevant aspects for ensuring a player returns to play under optimum conditions. Finally, the RONDO-TEST may be validated for any given injury, like the ACL rupture due to the injury burden for athletes.

## Figures and Tables

**Figure 1 jfmk-10-00003-f001:**
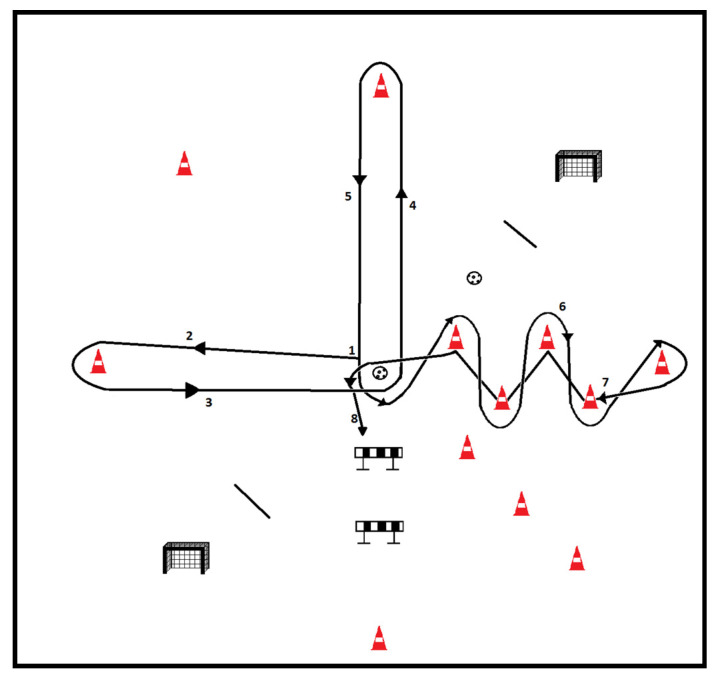
RONDO-TEST draft. The figure shows an example of the preliminary version of the RONDO-TEST design that was finally changed.

**Figure 2 jfmk-10-00003-f002:**
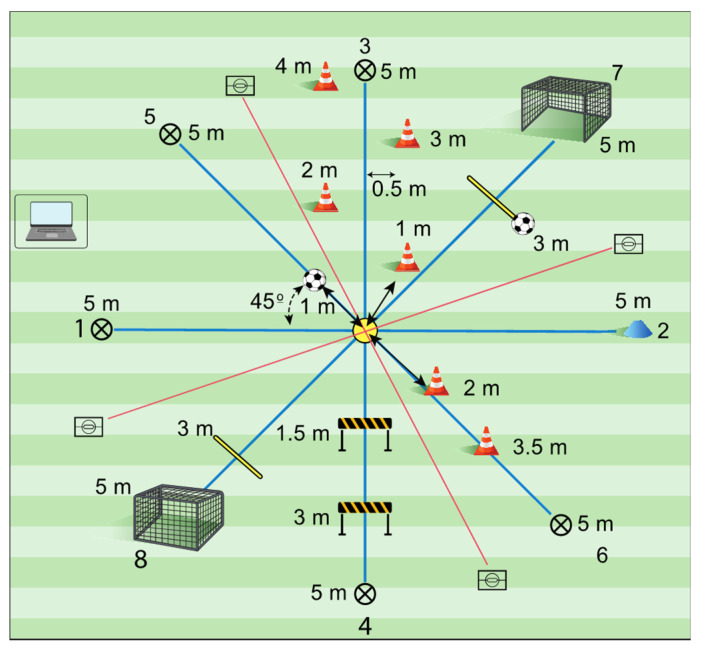
RONDO-TEST detailed procedures. Starting point: The athlete stands 1 m from the center of the rondo above the line of Sector 2. They freely begin the attempt, running forward in the direction of Sector 1. The time starts when they break the photoelectric beam. Additionally, an acoustic signal is emitted to confirm that the beam was adequately broken every time they pass through the center. Sector 1 (Forward/Backward): Run forward to the pole and go backward around it./Run backward to the center. Sector 2: Move sideways to the ground marker and step on it/go back to the center. Sector 3: Go around the outside of each cone and the pole/go back while doing a lateral support on each cone. Sector 4: Jump over the 2 mini-hurdles and receive the ball with any limb. Go around the pole/jump over the first mini-hurdle and receive the ball with the same limb. Jump over the second mini-hurdle and land on the other limb. Sector 5: Take the ball and dribble linearly with the same limb. Go around the pole/dribble linearly with the other limb. Sector 6: Dribble with 1 limb and go around the cones and the pole/dribble with the other limb and go around the cones. Sector 7: Dribble freely up to the 3 m line from the mini-goal. Kick the ball into the goal with 1 limb/take the second ball located at the 3 m line from the mini-goal and dribble freely while returning to the center. Sector 8: Dribble freely until the next 3 m line from the second mini-goal. Kick the ball into the goal/go back to the center as fast as possible to finish the test.

**Table 1 jfmk-10-00003-t001:** RONDO-TEST skills and related traditional tests.

Skill	Related Test
Speeding forward and backward	T-Test
	Illinois Agility Test
Moving sideways	T-Test
Side cutting and side jumping	Illinois Agility Test
	Triple-Hop Test
Two-foot and one-foot jumping	Triple-Hop Test and Hop Test
Ball linear dribbling	Shuttle Test
Ball dribbling slalom course	
Ball dribbling and kicking/passing	

**Table 2 jfmk-10-00003-t002:** Aiken’s V coefficient values and confidence intervals for the appropriateness of each item.

Item	1st Round			2nd Round		
	V	95% CI Lower Limit	95% CI Upper Limit	V	95% CI Lower Limit	95% CI Upper Limit
A1	0.31	0.18	0.49	0.44	0.28	0.61
B2	0.06	0.02	0.20	1.00	0.89	1.00
C3	0.53	0.36	0.69	1.00	0.89	1.00
D4 Sector 1	0.66	0.48	0.80	0.91	0.76	0.97
D5 Sector 2	0.56	0.39	0.72	0.91	0.76	0.97
D6 Sector 3	0.59	0.42	0.74	0.91	0.76	0.97
D7 Sector 4	0.63	0.45	0.77	0.91	0.76	0.97
D8 Sector 5	0.59	0.42	0.74	1.00	0.89	1.00
D9 Sector 6	0.59	0.42	0.74	1.00	0.89	1.00
D10 Sector 7	0.44	0.28	0.61	0.56	0.39	0.72
D11 Sector 8	0.47	0.31	0.64	0.56	0.39	0.72
E12 Total time	1.00	0.89	1.00	1.00	0.89	1.00
E13 Time per each Sector	1.00	0.89	1.00	1.00	0.89	1.00
E14 Subtotal time grouping skills without ball	1.00	0.89	1.00	1.00	0.89	1.00
E15 Subtotal time grouping skills with ball	1.00	0.89	1.00	1.00	0.89	1.00
F16	1.00	0.89	1.00	1.00	0.89	1.00

A1, clarity of instructions to perform the test; B2, order of execution and global development; C3, material organization; D4–D11, sector description and appropriateness of the skill; E12–E15, relevance to measure total and partial times; F16, appropriateness and feasibility of using photoelectric beam and acoustic signal to record time.

## Data Availability

The original contributions presented in the study are included in the article/Appendix A. Further inquiries can be directed to the corresponding authors.

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
