# Peer review of "Content Validity of a New Soccer (Football) Return-to-Play Test: The RONDO-TEST"

_jfmk, 2024, doi:10.3390/jfmk10010003_

Round 1
Reviewer 1 Report
Comments and Suggestions for Authors
the definition of RTP is not described, is it a return to training soccer or an ability to play competitively?
In the Delphi procedure, there are 2 soccer teams therapist whose specialties are unknown. Why not include sports doctors or rehabilitation physicians?
The RTP is so different depending on the type of injury, why validate the rondo-test for all pathologies rather than just one injury?
For RTP, scientific data highlight chaos, fatigue and neurocognitive management, so why not integrate them into the rondo-test?
Author Response
RESPONSE LETTER TO THE REVIEWER 1
Dear editor and reviewer, thank you very much for taking the time to review this manuscript. We made some changes as suggested, and provide answers to the topics that were not clear enough. Please, find the detailed responses below and the corresponding revisions highlighted in the re-submitted files.
Comments 1: the definition of RTP is not described, is it a return to training soccer or an ability to play competitively?
Response 1: Thank you for pointing this out. We added a sentence to describe RTP (lines 51-54)
“It should be highlighted that RTP as a process in clinical practice may refer from full return without restrictions to allow exercising, but the final decision after the process includes medical clearance of the athlete for full participation without restriction”
C: In the Delphi procedure, there are 2 soccer teams therapist whose specialties are unknown. Why not include sports doctors or rehabilitation physicians?
R: We already described that the Delphi process included sports doctors / physicians, as mentioned in lines 103-104. However, we did not describe what sort of therapists, and now we added it in line 104-105
“sport physiotherapists”.
C: The RTP is so different depending on the type of injury, why validate the rondo-test for all pathologies rather than just one injury?
R: Thank you to arise the question. The aim of the study was to design and assess the content validity of the test, and we already suggested that future studies could address specific populations.
However, to provide a clear answer to your question, we add the following sentence in lines 217-220
“RTP is different depending on the type of injury (ie., injury mechanism, tissue being affected) and RONDO-TEST can lead to obtain basal values prior the onset of injury and, hence, monitor basic and soccer-specific motor patterns alongside the RTP process thanks to the RONDO-TEST being segmented”
Also, in lines 245-246:
“Finally, RONDO-TEST may be validated for any given injury, like the ACL rupture due to the injury burden for athletes.”
C: For RTP, scientific data highlight chaos, fatigue and neurocognitive management, so why not integrate them into the rondo-test?
R: We recognise that this may be a limitation when trying to simulate real conditions. However, we highlight the importance of avoiding confounding variables between the RONDO-TEST time of execution and the healing process. See lines 213-217.
“RONDO-TEST procedures received consensus despite not including decision-making during the execution (eg., visual stimuli to decide the segment to perform), as it was designed originally. The reason for it is to administer the RONDO-TEST with the very same conditions, so the main outcomes (total and partial times) will not be affected by decision-making as a confounding variable”.
Reviewer 2 Report
Comments and Suggestions for Authors
The aim was to assess the content validity of a new field test on general and soccer-specific motor skills before return-to-play- This is a well-structured study from a scientific point of view, presenting the innocence of validating a new test to improve previous validations. However, some sections need minor improvements.
Abstract: Should clarify which qualitative magnitude is associated with V = 1.
Introduction: It should clarify the validity, reliability and internal consistency values of the other tests (Hop Test, Triple Hop, Test, T-Test, Illinois Test, Multi-Stage Fitness Test, Cooper Test, Shuttle Test, Balance Test). And indeed, why and what the RONDO test will really bring.
Materials and methods: The RONDO-TEST skills test has actually been constructed in a very interesting way. It is important to understand the logic behind the construction of table 1 and what it will evaluate differently to the studies that already exist.
Results: Figure 1 and 2 should be framed within the methodological procedures of the previous section. The results section should present and describe the figures in table 2.
Discussion: The writing is analytical and fits in well with the type of study, but it was important to expand on the practical applications, limitations of the study and future prospects. In addition, the relationship reported for LESS and ACL injury seems to me to be poorly substantiated with these results, at the very least a future validation of this test for this purpose could be attempted.
Conclusions: Although the last sentence could be about future prospects.
Author Response
Dear editor and reviewer, thank you very much for taking the time to review this manuscript. We made some changes as suggested, and provide answers to the topics that were not clear enough. Please, find the detailed responses below and the corresponding revisions highlighted in the re-submitted files.
Comments 1: Abstract: Should clarify which qualitative magnitude is associated with V = 1.
Response 1: Thank you for pointing this out. We added the description in the Abstract, line 27:
“maximum acceptance”.
C: Introduction: It should clarify the validity, reliability and internal consistency values of the other tests (Hop Test, Triple Hop, Test, T-Test, Illinois Test, Multi-Stage Fitness Test, Cooper Test, Shuttle Test, Balance Test). And indeed, why and what the RONDO test will really bring.
R: In relation to the validity of other tests, we included references in lines 62-24:
“Validation studies exist for the beforementioned tests -hop testing [14], Illinois tests [15], Shuttle tests [16] and Balance tests [17]- but the prognostic capacity following serious injuries is yet unclear [14].”
By regards of what RONDO-TEST brings, we already describe in line 66-69 that current tests do not evaluate comprehensively or are unfeasible. The aim of our study indicates that the RONDO includes, then, general and soccer-specific motor skills
C: Materials and methods: The RONDO-TEST skills test has actually been constructed in a very interesting way. It is important to understand the logic behind the construction of table 1 and what it will evaluate differently to the studies that already exist.
R: We add a highlight that the RONDO is a comprehensive test, as shown in lines 90-92:
“Therefore, the RONDO-TEST provides a more comprehensive evaluation of performance in the context of football unlike tests that assess a single sports action in an isolated and decontextualized environment”
C: Results: Figure 1 and 2 should be framed within the methodological procedures of the previous section. The results section should present and describe the figures in table 2.
R: We moved Fig 1 to Material and Methods (see lines 94-97), but we kept Fig 2 in Results section because the resulting RONDO-TEST is indeed a result of the Delphi methodology. We added a reference to a Supplementary Material with a full description of the RONDO-TEST (lines 139-140).
C: Discussion: The writing is analytical and fits in well with the type of study, but it was important to expand on the practical applications, limitations of the study and future prospects. In addition, the relationship reported for LESS and ACL injury seems to me to be poorly substantiated with these results, at the very least a future validation of this test for this purpose could be attempted.
R: We add the following sentence in lines 217-220
“RTP is different depending on the type of injury (ie., injury mechanism, tissue being affected) and RONDO-TEST can lead to obtain basal values prior the onset of injury and, hence, monitor basic and soccer-specific motor patterns alongside the RTP process thanks to the RONDO-TEST being segmented”
Also, in lines 245-246
“Finally, RONDO-TEST may be validated for any given injury, like the ACL rupture due to the injury burden for athletes.”
C: Conclusions: Although the last sentence could be about future prospects.
R: Answered also for the previous comment.
Reviewer 3 Report
Comments and Suggestions for Authors
First of all, I would like to thank the editors of the journal for giving me the opportunity to review this article for the journal.
With regard to the article, I think it is a very interesting piece of research. Proposing and empirically demonstrating a new physical test related to football, one of the most played sports in the world, is very useful and relevant.
However, I would like to make a number of recommendations for this work to be published in the journal:
- In the introduction, it is recommended that a current review be made of research that has carried out similar measurements in recent years. In other words, the aim is to find out what has been written in the literature about tests, circuits or physical tests similar to those carried out by the authors.
- In the methodology section, it would be advisable to give a brief explanation of the objectives of this test. As the authors explain in table 1, different skills or abilities are measured, similar to other tests. These tests only measure one specific skill or ability. Do you think it would be useful to measure all of them in one test? As can be seen in the video they have uploaded in the supplementary information, the player who performs it, for example, does not reach the required limits on several occasions.
- On the other hand, has the test been tried out with any football teams, of what age and gender, and what competitive level? It would be very interesting to be able to do this in order to analyse the viability of the test, not only at a statistical level.
- As in the introduction, it is recommended that, in the discussion, a literature search of recent years be carried out and the results obtained be compared. As we have not conducted a test with players, the discussion is very simple.
Author Response
Dear editor and reviewer, thank you very much for taking the time to review this manuscript. We made some changes as suggested, and provide answers to the topics that were not clear enough. Please, find the detailed responses below and the corresponding revisions highlighted in the re-submitted files.
Comments 1: - In the introduction, it is recommended that a current review be made of research that has carried out similar measurements in recent years. In other words, the aim is to find out what has been written in the literature about tests, circuits or physical tests similar to those carried out by the authors.
Response 1: Thank you for pointing this out. We included new references for the tests that were already mentioned in lines 62-24:
“Validation studies exist for the beforementioned tests -hop testing [14], Illinois tests [15], Shuttle tests [16] and Balance tests [17]- but the prognostic capacity following serious injuries is yet unclear [14].”
We also included a new reference of a recent article published in the same journal, with the limitation of being likely unfeasible for modest sport clubs (see lines 69-70), in contrast with RONDO TEST:
“For example, validity studies of laboratory testing [20] may not feasible broadly (eg., grassroot or regional sport clubs).”
C: In the methodology section, it would be advisable to give a brief explanation of the objectives of this test. As the authors explain in table 1, different skills or abilities are measured, similar to other tests. These tests only measure one specific skill or ability. Do you think it would be useful to measure all of them in one test? As can be seen in the video they have uploaded in the supplementary information, the player who performs it, for example, does not reach the required limits on several occasions.
R: Thank you for the comments. By regards of the objectives of the test, we include a sentence pointing out that it should be done as fast as possible (line 89):
“The RONDO-TEST includes four 10-meter lines which cross over at their mid-point, resulting in eight 5-meter sectors that have to be executed as fast as possible”.
In relation to the comment of the usefulness to measure many skills in one test, we already explained in the Introduction section (lines 54-59) and, specially, in lines 64-67 and lines 76-79:
“Medical and injury risk factors to consider before an RTP decision include complete suppression of pain, repair of injured tissues, restoration of metabolic pathways that guarantee energy supply, elimination of muscle and joint disbalance due to inactivity, and re-education of general and soccer-specific movement patterns [11]. These risk factors are essential to consider because the risk of re-injury increases when rehabilitation is inadequate or the RTP is premature [12,13] (…) Additionally, Lalín and Peirau [8] suggest that functional tests assessing motor skills and specific soccer patterns during the re-adaptation and re-training process permit the assessment and identification of injury risk factors, aiding the final decision of RTP”
Third, in relation to the video, the player does not step on the cone one time only. However, we agree with you that the video recording aids at determining the reasons of poor or incorrect execution. Therefore, we added in Discussion (lines 220-222):
“Qualitative information may be observed if the RONDO-TEST execution is videorecorded, as for analysing possible reasons of poor execution”
C: On the other hand, has the test been tried out with any football teams, of what age and gender, and what competitive level? It would be very interesting to be able to do this in order to analyse the viability of the test, not only at a statistical level.
R: No. The aim of the study was to design and assess the content validity of the test. However, we are glad that you consider that it may be interesting to analyse the administration of the test to different populations. We already suggested that future studies could address the issue (lines 241-246):
“Also, reference values from different populations (according to sex, age, level of performance) may provide the assessment of technical functional skills (actions) and condition-related abilities (the ability to repeat the test, fatigue curve, etc.), which are relevant aspects for ensuring a player returns to play under optimum conditions. Finally, RONDO-TEST may be validated for any given injury, like the ACL rupture due to the injury burden for athletes.”
C: As in the introduction, it is recommended that, in the discussion, a literature search of recent years be carried out and the results obtained be compared. As we have not conducted a test with players, the discussion is very simple.
R: To our knowledge, there is not any test that assess comprehensively basic and soccer-specific motor skills, as we pointed out in the Introduction, so there is lack of data to compare to.
Round 2
Reviewer 3 Report
Comments and Suggestions for Authors
First of all, I would like to thank the authors for their efforts to improve this research work.
The authors have made most of the changes required in the first revision. However, I would like to stress that a more extensive review of the scientific literature in this area is missing, although the authors state that it is the first test of this kind, as well as testing the validity of the test with a sample of football players.
How can we test the validity of the questionnaire without having put it into practice? What is the target population? Is it valid for all types of football players?
Author Response
Dear reviewer, thank you again for your time and suggestions. We apologise of not providing enough clarity on our previous answers, and hope that our response now is addressing the issues. Again, find the detailed responses below and the corresponding revisions highlighted in the re-submitted files.
Comments: The authors have made most of the changes required in the first revision. However, I would like to stress that a more extensive review of the scientific literature in this area is missing, although the authors state that it is the first test of this kind, as well as testing the validity of the test with a sample of football players.
Response: We proceeded to search recent evidence on the content validity of RTP tests in football (soccer), and updated the Introduction section (see lines 65-67, and 69-70). The results from the search on PubMed found the following results (as of 20th December 2024):
Equation: [“content validity” AND (football OR soccer) AND (“return-to-play” OR RTP)]: 0 results.
Equation: [validity AND (football OR soccer) AND (“return-to-play” OR RTP)]: 57
The following issues were found relevant and, therefore included in the manuscript: an ecological perspective to assess RTP outcomes after ACL injury (Forelli F, et al, 2023) , and the validity of RTP programmes after muscle injuries in soccer (Jiménez-Rubio S etal, 2019, 2021).
Other reserarch related with the aim of our study was also found, but they were out of the scope:
- The aim was to find common criteria of items to support RTP decisions, but they do not suggest any specific test to evaluate the decisions (Zambaldi M, Beasley I, Rushton A. Return to play criteria after hamstring muscle injury in professional football: a Delphi consensus study. Br J Sports Med. 2017;51(16):1221-1226. doi:10.1136/bjsports-2016-097131)
- The aim of the study was not related specifically to the RTP process. Authors suggest in the end that the running kinematic’s variables may be useful for RTP. It is not a comprehensive test (Alonso-Callejo A, García-Unanue J, Guitart-Trench M, Majano C, Gallardo L, Felipe JL. Validity and Reliability of the Acceleration-Speed Profile for Assessing Running Kinematics' Variables Derived From the Force-Velocity Profile in Professional Soccer Players. J Strength Cond Res. 2024;38(3):563-570. doi:10.1519/JSC.0000000000004637)
- These studies evaluated the psychometric properties of questionnaires. It did not include fitness testing (Bley JA, Master H, Huston LJ, et al. Return to Sports After Anterior Cruciate Ligament Reconstruction: Validity and Reliability of the SPORTS Score at 6 and 12 Months. Orthop J Sports Med. 2022;10(6):23259671221098436. Published 2022 Jun 8. doi:10.1177/23259671221098436; Dunlop G, Ivarsson A, Andersen TE, et al. Examination of the validity of the Injury-Psychological Readiness to Return to Sport (I-PRRS) scale in male professional football players: A worldwide study of 29 professional teams. J Sports Sci. 2023;41(21):1906-1914. doi:10.1080/02640414.2024.2307764; Sala-Barat E, Álvarez-Díaz P, Alentorn-Geli E, Webster KE, Cugat R, Tomás-Sabado J. Translation, cross-cultural adaptation, validation, and measurement properties of the Spanish version of the anterior cruciate ligament-return to sport after injury (ACL-RSI-Sp) scale. Knee Surg Sports Traumatol Arthrosc. 2020;28(3):833-839. doi:10.1007/s00167-019-05517-z)
- A Delphi study suggests a battery of tests for upper limb RTP, not a single test that assesses comprehensively football actions (Kurz E, Bloch H, Buchholz I, et al. Assessment of return to play after an acute shoulder injury: protocol for an explorative prospective observational German multicentre study. BMJ Open. 2023;13(2):e067073. Published 2023 Feb 3. doi:10.1136/bmjopen-2022-067073)
- It did not evaluate the validity of any test. But a secondary reference was found that was, indeed, included in our Introduction section: Estévez-Rodríguez JL, Rivilla-García J, Jiménez-Rubio S. Is It Possible to Improve Performance in Competition After an Adductor Longus Injury in Professional Football Players?. J Sport Rehabil. 2024;33(8):663-667. Published 2024 Sep 6. doi:10.1123/jsr.2024-0028
Comment: How can we test the validity of the questionnaire without having put it into practice? What is the target population? Is it valid for all types of football players?
Response: We reworded a sentence in Conclusions section to underpin the usefulness of objective measurement rather than injuries. (Lines 314-315: This test may provide a feasible and simple way to evaluate objectively soccer motor skills and, potentially, the readiness of soccer players to participate in team trainings after an injury and, therefore, lowering the risk of re-injury because of premature return-to-play)
We figured out that it has been a typing mistake that the comment refers to the validity of a questionnaire, which was not to be validated. The questionnaire we used was to proceed with the Delphi process.
However, understanding that the reviewer’s comment referred to the RONDO-TEST to be validated, we focused the goal of our study to test the content validity, that is, the extent to which a measure thoroughly and appropriately assesses the skills or characteristics it is intended to measure (https://www.sciencedirect.com/topics/nursing-and-health-professions/content-validity), following a Delphi process. It is a previous step to evaluate the reliability, which is already suggested to do in future studies in which the target population (profile of players) should be clearly described (eg., sex, level of performance, healthy or with a specific injury).

Round 3
Reviewer 3 Report
Comments and Suggestions for Authors
Authors have made the requested changes.